# Differential conditioning produces merged long-term memory in *Drosophila*

**Bohan Zhao[1,2†], Jiameng Sun[1,2†], Qian Li[1,2], Yi Zhong[1,2]\***

[1]School of Life Sciences, IDG/McGovern Institute for Brain Research, and MOE Key Laboratory for Protein Science, Tsinghua University, Beijing, China; [2]Tsinghua-Peking Center for Life Sciences, Beijing, China

**Abstract** Multiple spaced trials of aversive differential conditioning can produce two independent long-term memories (LTMs) of opposite valence. One is an aversive memory for avoiding the conditioned stimulus (CS+), and the other is a safety memory for approaching the non-conditioned stimulus (CS–). Here, we show that a single trial of aversive differential conditioning yields one merged LTM (mLTM) for avoiding both CS+ and CS–. Such mLTM can be detected after sequential exposures to the shock-paired CS+ and -unpaired CS–, and be retrieved by either CS+ or CS–. The formation of mLTM relies on triggering aversive-reinforcing dopaminergic neurons and subsequent new protein synthesis. Expressing mLTM involves αβ Kenyon cells and corresponding approach-directing mushroom body output neurons, in which similar-amplitude long-term depression of responses to CS+ and CS– seems to signal the mLTM. Our results suggest that animals can develop distinct strategies for occasional and repeated threatening experiences.

**\*For correspondence:**
zhongyithu@tsinghua.edu.cn

[†]These authors contributed equally to this work

**Competing interests:** The authors declare that no competing interests exist.

## Introduction

To survive in a complex environment, animals need to learn from threatening experiences to avoid potential dangers. From invertebrates to humans, aversive differential conditioning is widely used to study memories produced by threatening experiences (*Carew et al., 1983*; *Corches et al., 2019*; *Gentile et al., 1986*; *Schneider et al., 1999*; *Tully and Quinn, 1985*). After repetitive spaced trials of conditioning, animals form two complementary long-term memories (LTMs) of opposite valence (*Jacob and Waddell, 2020*; *Pollak et al., 2008*; *Pollak et al., 2010*), including the aversive memory to the conditioned stimulus (CS+) and the rewarding memory to the non-conditioned stimulus (CS–) (*Figure 1—figure supplement 1*). Such complementary LTMs result in enhanced long-lasting discrimination between CS+ and CS– through guiding avoidance to CS+ and approach to CS–. However, it remains unclear whether and how occasional threatening experiences, such as single-trial conditionings, would induce long-lasting changes in future escape behavior.

From invertebrates to humans, experience-dependent long-lasting behavioral modifications mainly rely on the formation of LTMs (*Kandel et al., 2014*). In *Drosophila*, there are at least two categories of aversive olfactory LTMs that last for more than 7 days. One is the spaced training-induced LTM that can be observed after repetitive spaced training with inter-trial rests (multiple trials with a 15 min rest interval between each), but not after either single-trial training or repetitive massed training without interval (*Tully et al., 1994*). Forming such aversive LTM requires new protein synthesis and the paired posterior lateral 1 (PPL1) cluster of dopaminergic neurons (DANs) to depress the connection between odor-activated Kenyon cells (KCs) in the mushroom body (MB) αβ lobe and downstream α2sc (MB-V2) MB output neurons (MBONs) (*Aso et al., 2014a*; *Aso et al., 2014b*; *Cognigni et al., 2018*; *Modi et al., 2020*; *Pascual and Préat, 2001*; *Plaçais et al., 2012*). The other is a recently reported context-dependent LTM that forms after single-trial training, which does not require protein synthesis-dependent consolidation (*Zhao et al., 2019*). The expression of context-

dependent LTM relies on multisensory integration in the lateral horn and is not affected by blocking KCs. However, all these observations derived from the same design principle that evaluates memory performance through testing the discrimination between CS+ and CS– (*Tully and Quinn, 1985*). Thus, direct responses to CS+ and CS– have been largely overlooked.

In the current study, we introduced a third-odor test, in which flies were given a choice between either CS+ and a novel odor, or CS– and a novel odor. We therefore identified that the single-trial differential conditioning produces a merged LTM (mLTM) guiding avoidances of both CS+ and CS– for several days after training. The encoding and expression of such mLTM involve new protein synthesis, PPL1 DANs, αβ KCs, and α2sc MBONs. These findings suggest that animals utilize distinct escape strategies for facing occasional and repeated dangers.

## Results

### Single-trial training produces aversive LTMs to both CS+ and CS–

We first plotted the memory retention curve at various time points after training (*Figure 1A*). Consistent with previous reports (*Davis and Zhong, 2017*; *Shuai et al., 2010*; *Tully et al., 1994*), flies exposed to single-trial conditioning rapidly lost their ability to discriminate CS+ and CS– within 1 day. However, when trained flies were instead tested between either CS+ and a novel odor, or CS– and a novel odor, they gradually exhibited significant avoidances of both CS+ and CS– from 12 hr to more than 7 days after training (*Figure 1B*). Consistently, testing CS+ or CS– versus air also revealed training-induced long-lasting avoidances of CS+ and CS– (*Figure 1—figure supplement 2*).

Based on these observations, we wondered whether such aversive LTMs require new protein synthesis to consolidate. Results show that such avoidances resulted from the formation of protein synthesis-dependent LTM$^{CS+}$ and LTM$^{CS–}$ because the administration of cycloheximide (CXM), a protein synthesis inhibitor, prevented the avoidances of both CS+ and CS– after 1 day (*Figure 1C*). The time course of the consolidation of such LTMs could be revealed by cold-shock treatment, which is known to abolish labile memories but leave consolidated LTM intact (*Krashes and Waddell, 2008*; *Li et al., 2016*; *Tully et al., 1994*). As shown in *Figure 1D*, it takes more than 3 hr for consolidation to be completed.

### The long-term avoidances of CS+ and CS– seem to be derived from the same mLTM

The above results prompted us to investigate whether LTM$^{CS+}$ and LTM$^{CS–}$ are based on two parallel LTMs of the same valence, or the same LTM that can be retrieved by either odor. Two lines of evidence suggested that LTM$^{CS+}$ and LTM$^{CS–}$ are derived from the same memory component.

First, LTM$^{CS+}$ and LTM$^{CS–}$ were both extinguished by the re-exposure to either CS+ or CS– alone (*Figure 2A*). Trained flies were re-exposed to three cycles of either CS+ or CS– for 1 min with 1 min inter-trial intervals immediately before testing. Such treatment significantly reduced both LTM$^{CS+}$ and LTM$^{CS–}$, as compared to those re-exposed to air, suggesting the LTM$^{CS+}$ and LTM$^{CS–}$ are both retrievable for either one of CS+ and CS–.

Second, LTM$^{CS+}$ and LTM$^{CS–}$ were both abolished when the temporal interval between CS+ and CS– during training was prolonged to more than 5 min (*Figure 2B*), suggesting that the formation of LTM$^{CS+}$ and LTM$^{CS–}$ depends on the contiguity of two shock-paired CS+ and CS– exposures, instead of temporally separated exposures. In addition, changing the sequence of exposure to CS+ and CS– did not affect either LTM$^{CS+}$ or LTM$^{CS–}$ (*Figure 2C*), but training with CS+ or CS– alone failed to induce the LTM$^{CS+}$ or LTM$^{CS–}$ (*Figure 2—figure supplement 1*).

Together, these results strongly suggest that, instead of two parallel LTMs, observed LTM$^{CS+}$ and LTM$^{CS–}$ are more likely to be derived from the same memory component. Since it was observable after two separated memory curves (CS+ and CS–) merged over time, we termed it as mLTM.

### The encoding of mLTM requires aversive-reinforcing DANs

Then we further investigated the neural network mechanisms underlying such mLTM. In *Drosophila*, associative olfactory memories are mainly encoded by the DANs (*Aso and Rubin, 2016*; *Claridge-Chang et al., 2009*; *Cognigni et al., 2018*) through alterations of the connections between KCs and MBONs (*Aso et al., 2014a*; *Aso et al., 2014b*; *Cognigni et al., 2018*; *Dubnau and Chiang, 2013*;

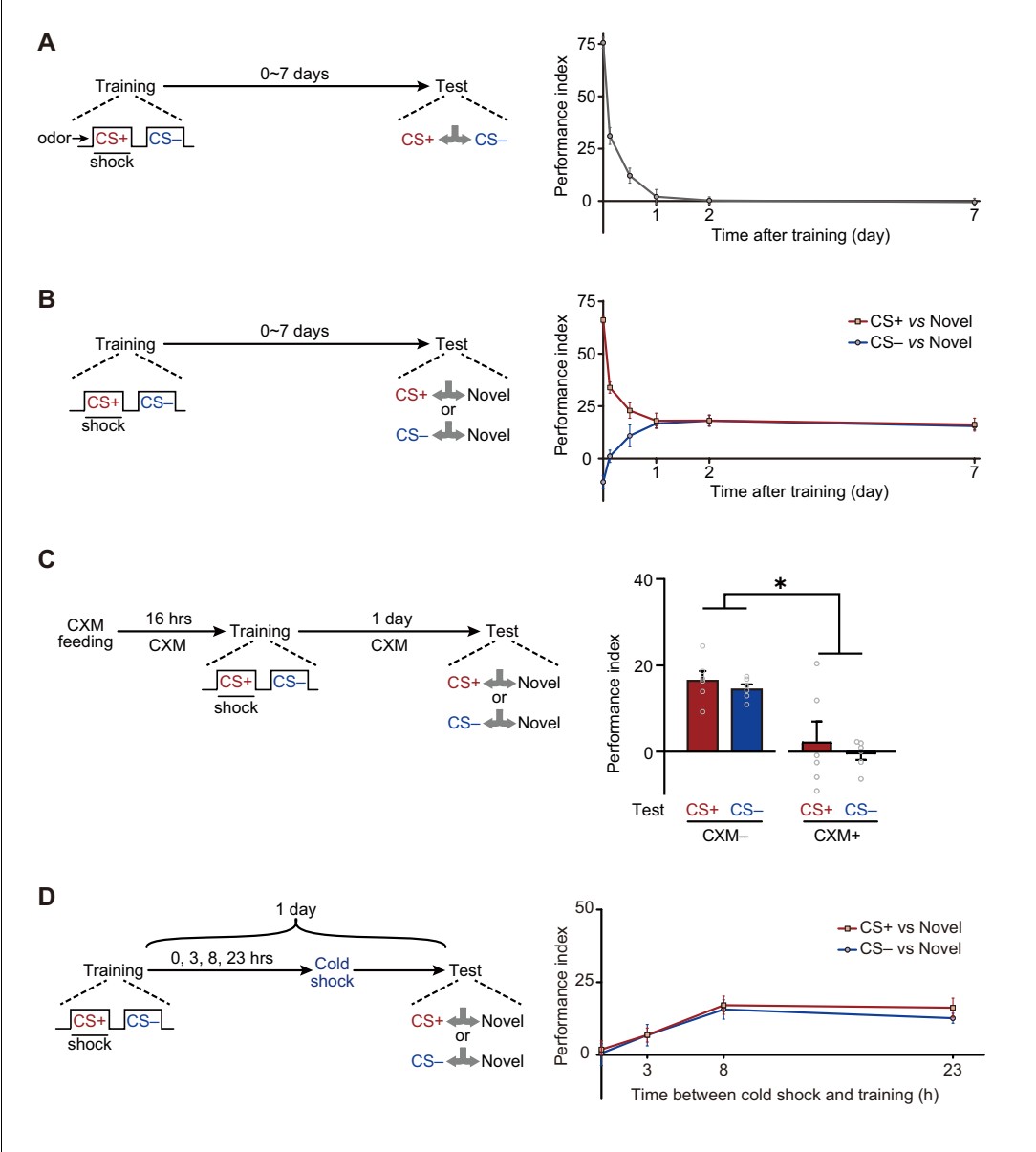

**Figure 1.** Long-term avoidances of both conditioned stimulus (CS+) and non-conditioned stimulus (CS–) induced by aversive differential conditioning. (A) Single-trial differential conditioning induced discriminative memory that was forgotten within 1 day (n = 6–8). (B) Conditioned flies exhibited long-lasting avoidances of both CS+ and CS– (n = 8–10). (C) Cycloheximide (CXM) treatment abolished 1 day avoidances of CS+ and CS– (n = 6). (D) The 1 day avoidances were tested after cold-shock anesthesia at different time points following training. Aversive memory to CS+ and CS– requires a multi-hour consolidation process (n = 6–8). All data shown are presented as mean ± SEM. *p < 0.05.

The online version of this article includes the following source data and figure supplement(s) for figure 1:

**Source data 1.** Behavioral data of each group in *Figure 1*.

**Figure supplement 1.** Repetitive spaced training forms complementary long-term memories (LTMs).

**Figure supplement 1—source data 1.** Behavioral data of each group in *Figure 1—figure supplement 1*.

**Figure supplement 2.** Single-trial differential conditioning increases conditioned stimulus (CS+) and non-conditioned stimulus (CS–) odor avoidances.

**Figure supplement 2—source data 1.** Behavioral data of each group in *Figure 1—figure supplement 2*.

*Modi et al., 2020*; *Pascual and Préat, 2001*; *Plaçais et al., 2012*). We therefore tested whether DANs involve in mLTM encoding by comparing 24 hr memory in control flies with that in flies whose synaptic outputs from different clusters of DANs were blocked during training. For this purpose, we expressed the dominant-negative temperature-sensitive *UAS-Shibire^{ts1}* (*Shi^{ts}*)-encoded dynamin

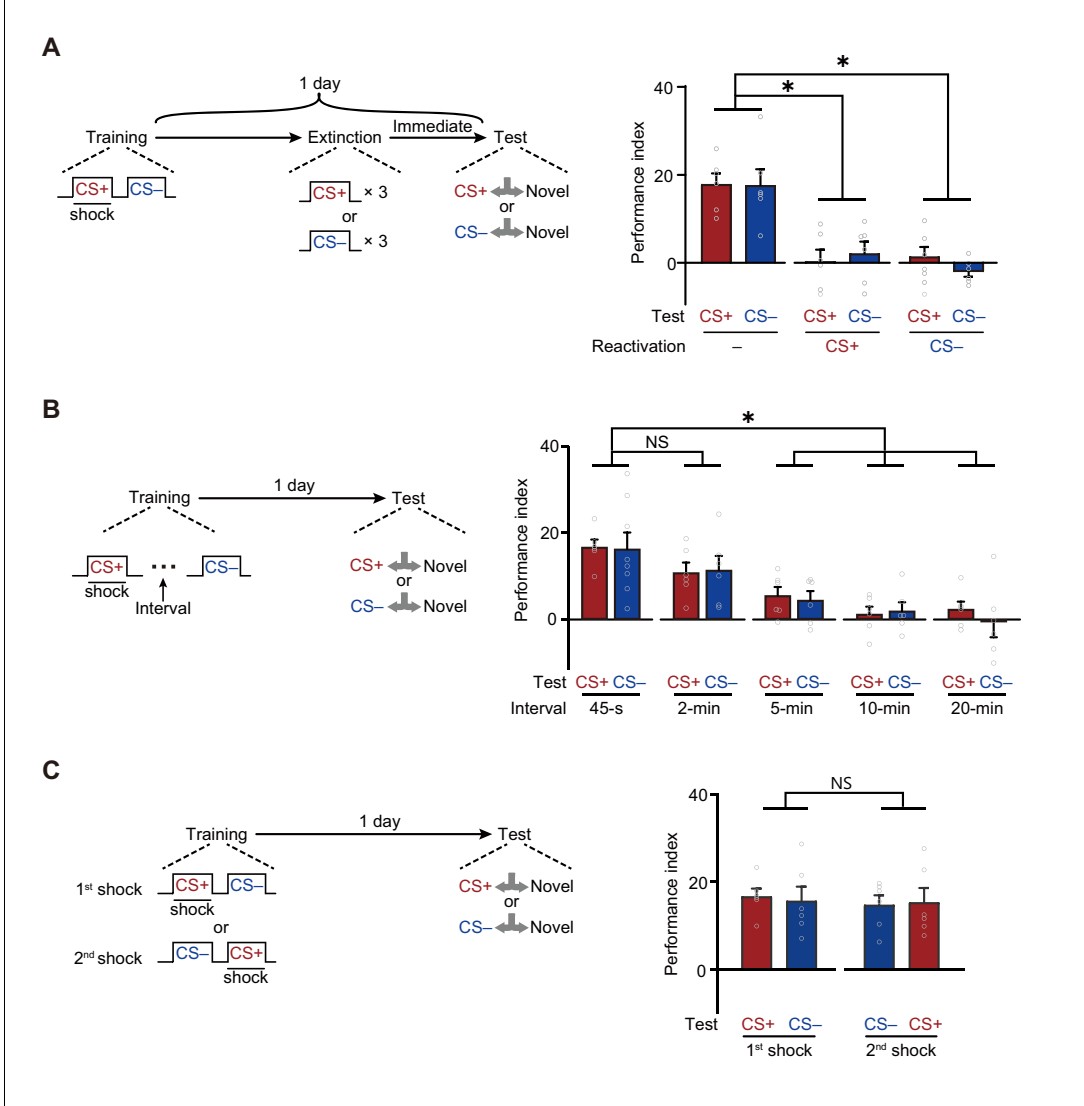

**Figure 2.** A merged long-term memory (mLTM) underlies the long-term avoidances of conditioned stimulus (CS+) and non-conditioned stimulus (CS–). (A) Three trials of re-exposure to either CS+ or CS– alone can impair both CS+ avoidance and CS– avoidance 1 day after training (n = 6). (B) Prolonging the inter-trial interval (ITI) between CS+ and CS– to more than 5 min significantly impaired 1 day avoidances (n = 6–8). (C) Changing the sequence of CS+ and CS– during training did not affect 1 day avoidances (n = 6). All data shown are presented as mean ± SEM. *p < 0.05. NS, non-significant.

The online version of this article includes the following source data and figure supplement(s) for figure 2:

**Source data 1.** Behavioral data of each group in *Figure 2*.

**Figure supplement 1.** Absolute trainings fail to induce 24 hr avoidances.

**Figure supplement 1—source data 1.** Behavioral data of each group in *Figure 2—figure supplement 1*.

---

(*Kitamoto, 2001*) driven by *R58E02-Gal4*, which labels the protocerebral anterior medial (PAM) cluster of DANs, and *R52H03-Gal4* and *TH-Gal4,* which both label PPL1 DANs (*Figure 3—figure supplement 1A,B and C*). We specifically blocked output from DANs during training by raising the temperature of flies from 23°C to 32°C. Flies were then returned to 23°C and later tested for 1 day memory. The results showed that blocking the release of neurotransmitter from PPL1 DANs, but not PAM DANs, during training impaired mLTM (*Figure 3A*), suggesting that the neuromodulation of PPL1 DANs is necessary for mLTM formation.

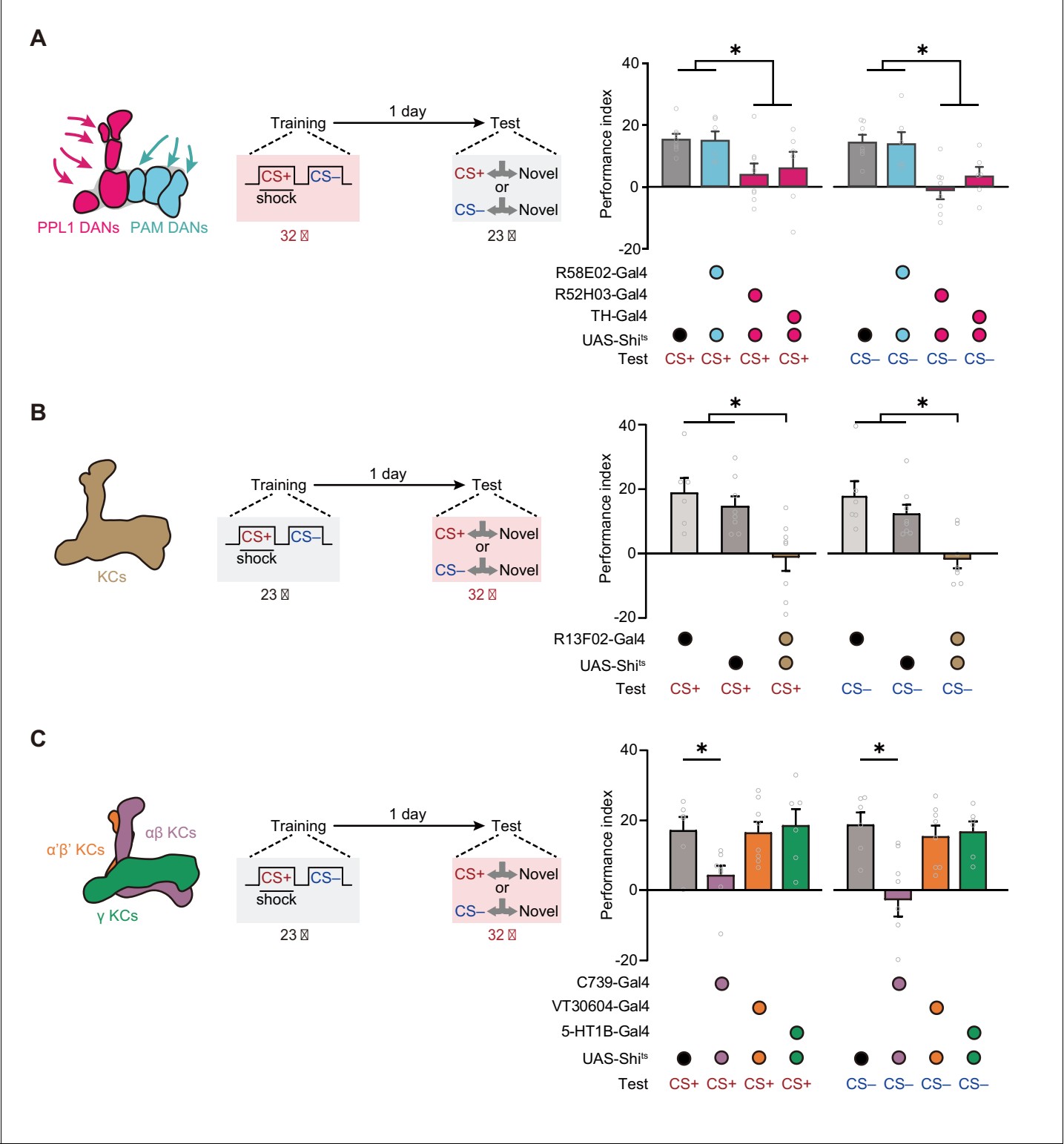

**Figure 3.** Expression of the merged long-term memory (mLTM) requires the αβ Kenyon cells (KCs). (**A**) Blocking paired posterior lateral 1 (PPL1) dopaminergic neurons (DANs) (*R52H03-Gal4* or *TH-Gal4*) but not protocerebral anterior medial (PAM) DANs (*R58E02-Gal4*) during training significantly impaired mLTM (n = 6–8). (**B**) Blocking mushroom body (MB) during testing using *R13F02-Gal4>UAS-Shi^ts* impaired the mLTM (n = 6–8). (**C**) The mLTM was impaired by blocking αβ KCs (*C739-Gal4*), but not that of α′β′ KCs (*VT30604-Gal4*) or γ KCs (*5-HT1B-Gal4*) (n = 6–8). All data shown are presented as mean ± SEM. *p < 0.05.

The online version of this article includes the following source data and figure supplement(s) for figure 3:

*Figure 3 continued on next page*

*Figure 3 continued*

**Source data 1.** Behavioral data of each group in *Figure 3*.

**Figure supplement 1.** The expression patterns of GAL4 lines used in *Figure 3*.

**Figure supplement 2.** Paired posterior lateral 1 (PPL1) and protocerebral anterior medial (PAM) dopaminergic neurons (DANs) play different roles in conditioned stimulus (CS+) and non-conditioned stimulus (CS–) memory encoding.

**Figure supplement 2—source data 1.** Behavioral data of each group in *Figure 3—figure supplement 2*.

### Retrieval of mLTM requires output of αβ KCs

We then examined the role of KCs in mLTM retrieval. We imposed the block by raising temperature to 32°C from 23°C, 15 min before the test. Inhibition of the synaptic output of all KCs (*R13F02-Gal4>UAS-Shi^ts*) through exposure to a restrictive temperature during the test abolished mLTM (*Figure 3B* and *Figure 3—figure supplement 1D*). We further tested the effects of blocking three distinct subgroups of KCs (*Figure 3C*), namely, those located within the MB αβ lobe (αβ KCs), which were labeled with *C739-Gal4*; those located within the α'β' lobe (α'β' KCs), which were labeled with *VT30604-Gal4*; and those located within the γ lobe (γ KCs), which were labeled with *5-HT1B-Gal4* (*Figure 3—figure supplement 1E-G*). The results showed that mLTM did not express when the synaptic output of αβ KCs was blocked, whereas inhibition of α'β' KCs and γ KCs had no significant effect (*Figure 3C*).

### The α2sc cluster of MBONs is required for mLTM expression

Then we further tried to identify which cluster of MBONs from αβ lobe of MB is required for mLTM. Prior studies on the structure of MB network have uncovered that the PPL1 DANs mainly integrate to the α2 and α3 compartments of the vertical α lobe of MB, and corresponding MBONs (α2sc and α3 clusters) have been reported to be important to aversive spaced training-induced LTM expression (*Aso et al., 2014a*; *Aso et al., 2014b*; *Bouzaiane et al., 2015*; *Dubnau and Chiang, 2013*; *Jacob and Waddell, 2020*; *Modi et al., 2020*; *Pascual and Préat, 2001*; *Plaçais et al., 2012*; *Schwaerzel et al., 2003*). We therefore attempted to block α2sc MBONs (*R71D08-Gal4*) and α3 MBONs (*G0239-Gal4*) during testing, respectively (*Figure 4—figure supplement 1A,B*). Results showed that mLTM expression was significantly impaired by blocking α2sc MBONs, but not α3 MBONs (*Figure 4A,B*). This conclusion was further strengthened by the result that inhibiting the neural activity of α2sc MBONs using heat-inducible expression of the potassium channel Kir2.1 decreased the mLTM retrieval (*Figure 4—figure supplement 2*). Thus, these results suggest that retrieving mLTM specifically relies on the output of α2sc MBONs.

### Differential conditioning induces long-lasting depression of CS+ and CS– odor-evoked responses in α2sc

We then imaged odor-evoked calcium responses in the dendritic field of α2sc MBONs to CS+ and CS– through expressing jGCaMP7f, a calcium-sensitive fluorescent protein (*Dana et al., 2019*), driven by *R71D08-Gal4*. To account for variance between different odors, the responses of trained flies were calibrated to the average responses of naïve flies to corresponding odors. Consistent with results of the behavioral assay, 1 day after training, odor-evoked calcium responses of α2sc MBONs to CS+ and CS– were both significantly reduced when compared to responses to the novel odor (*Figure 5A,B*). Moreover, such depression of CS+ and CS– odor-evoked responses can be abolished by prolonging the temporal interval between CS+ and CS– to 10 min during training (*Figure 5C*), which is consistent with the result of behavioral assay (*Figure 2B*). Interestingly, when we recorded the calcium responses immediately after training, the depressed responses to CS+ and CS– were observed, suggesting the mLTM should be encoded during training (*Figure 5—figure supplement 1*). Together, our data showed that the aversion to CS+ and CS– are both linked to the neural plasticity conferred by the same MBON.

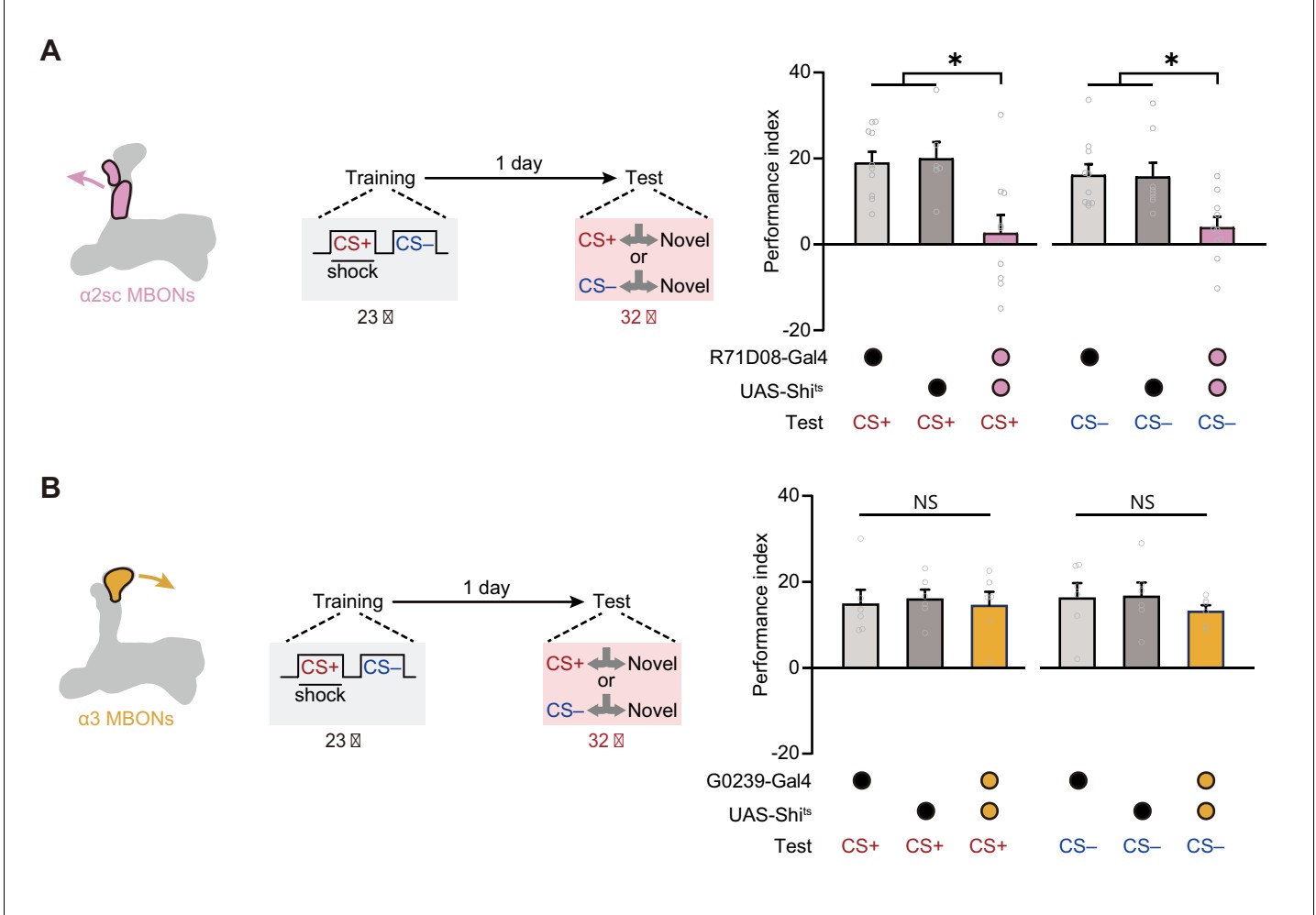

**Figure 4.** Expression of the merged long-term memory (mLTM) requires α2sc mushroom body output neurons (MBONs). (**A**) Blocking α2sc MBONs using *R71D08-Gal4>UAS-Shi^{ts}* impaired mLTM expression (n = 6–10). (**B**) Blocking α3 MBONs using *G0239-Gal4>UAS-Shi^{ts}* did not affect mLTM expression (n = 6). All data shown are presented as mean ± SEM. *p < 0.05. NS, non-significant.

The online version of this article includes the following source data and figure supplement(s) for figure 4:

**Source data 1.** Behavioral data of each group in *Figure 4*.

**Figure supplement 1.** The expression patterns of GAL4 lines used in *Figure 4*.

**Figure supplement 2.** Retrieving the merged long-term memory (mLTM) relies on the neural activity of α2sc mushroom body output neurons (MBONs).

**Figure supplement 2—source data 1.** Behavioral data of each group in *Figure 4—figure supplement 2*.

## Discussion

In the current study, the use of third-odor test leads to a conclusion that single-trial training produces an mLTM for guiding flies to avoid both CS+ and CS– for more than 7 days. Three categories of evidence in support of this conclusion are outlined below.

First, throughout our study, the amplitudes of long-term avoidances of CS+ and CS– are always at a similar level under various conditions, including pharmacological treatment (*Figure 1C*), cold-shock treatment (*Figure 1D*), odor re-exposure (*Figure 2A*), paradigm alteration (*Figure 2B and C*), and neural circuitry manipulations (*Figures 3* and *4*). Second, re-exposure to either one of CS+ and CS– alone can extinguish both CS+ avoidance and CS– avoidance (*Figure 2A*). Third, the long-term avoidances of CS+ and CS– can be recorded as the depression of odor-evoked responses in the same α2sc MBONs (*Figure 4*), meanwhile, CS+ avoidance and CS– avoidance both involve the same PPL1 DANs, αβ KCs, and α2sc MBONs (*Figures 3* and *4A*). Thus, CS+ avoidance and CS– avoidance

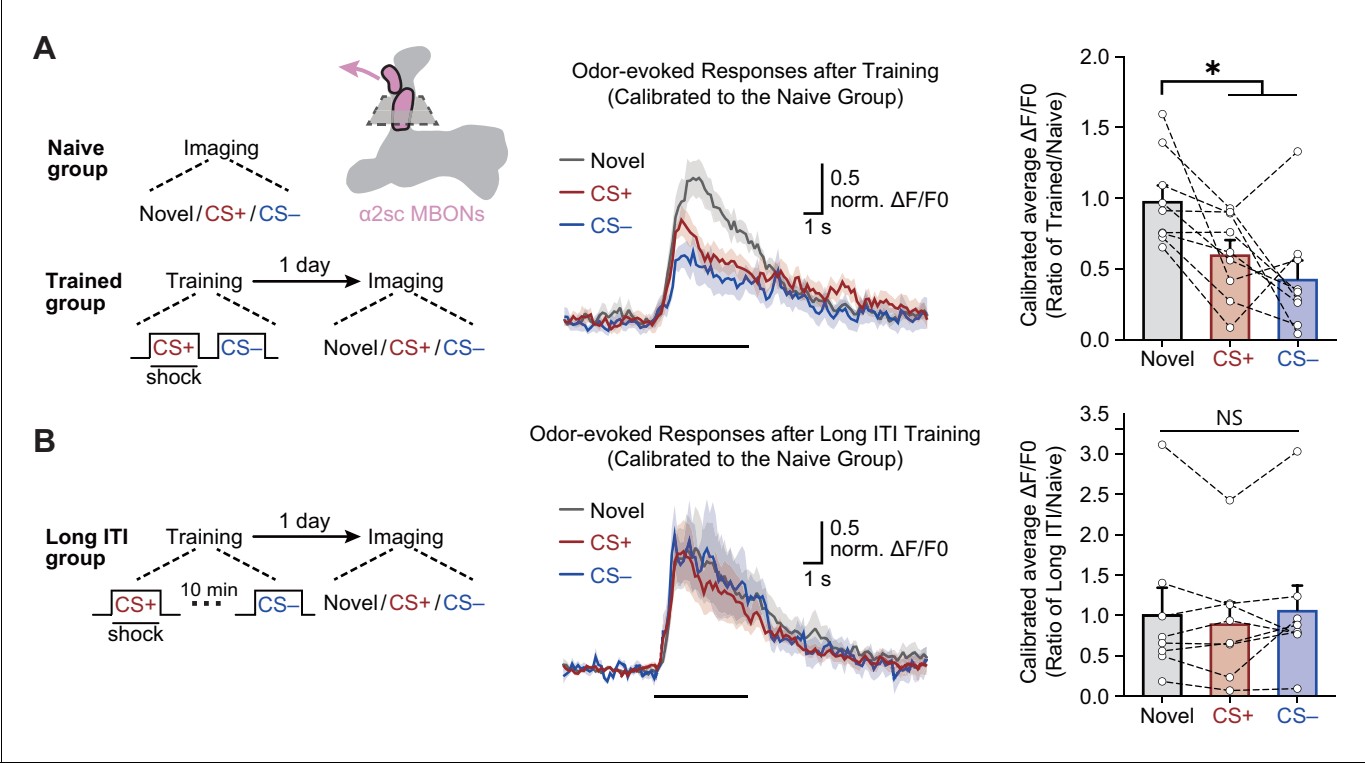

**Figure 5.** The merged long-term memory (mLTM) can be recorded as depressed odor-evoked responses in α2sc mushroom body output neurons (MBONs). (**A**) Left: training and imaging protocols. The imaging plane for α2sc MBONs is shown. The novel odor-evoked response, conditioned stimulus (CS+) odor-evoked response, and non-conditioned stimulus (CS–) odor-evoked response were calibrated to the average responses of corresponding odors in naïve flies. Right: the mLTM can be recorded as depressed odor-specific responses in α2sc MBONs (n = 9). (**B**) Prolonging the inter-trial interval (ITI) between CS+ and CS– during training abolished the depressed odor-specific responses (n = 8). All data shown are presented as mean ± SEM. *p < 0.05. NS, non-significant.

The online version of this article includes the following source data and figure supplement(s) for figure 5:

**Source data 1.** Calcium imaging data of each group in *Figure 5*.

**Figure supplement 1.** The depression of odor-evoked responses in α2sc mushroom body output neurons (MBONs) can be observed immediately after training.

**Figure supplement 1—source data 1.** Calcium imaging data of each group in *Figure 5—figure supplement 1*.

derive from the same aversive mLTM, instead of two parallel LTMs of the same valence. The significance of these findings is further discussed below.

Combining with a recent report that uses a similar third-odor test to dissect LTMs induced by multi-trial spaced training (*Jacob and Waddell, 2020*), we are led to conclude that spaced multi-trial aversive differential conditioning produces two independent LTMs of opposite valence for avoiding CS+ and approaching CS–, whereas single-trial aversive differential conditioning yields one mLTM that guides avoidances of both CS+ and CS–. Thus, animals can develop distinct escape strategies for different categories of dangers. When the same dangerous situation has been experienced repeatedly, animals would remember the detailed information to guide behavior in the next similar situation. However, when the dangerous event has only been experienced occasionally, animals would choose to avoid all potentially dangerous cues as a more reserved survival strategy.

Moreover, the differences between single-trial training-induced mLTM and multi-trial training-induced complementary LTMs lead us to ask: how these differences are induced by different training sessions? *Jacob and Waddell, 2020* reported that multi-trial spaced training induces depressed responses to CS+ in α2sc MBONs and α3 MBONs are required for aversive LTM to CS+, whereas the modulated responses to CS– in β'2mp MBONs and γ3, γ3β'1 MBONs appears to be responsible for the safety memory to CS–. In contrast, here we found that single-trial training is sufficient to induce the depressed responses to both CS+ and CS– in α2sc MBONs. Therefore, our results

suggest a lower threshold and specificity of the plasticity between KCs-α2sc MBONs, compared to KCs-α3 MBONs, KCs-β'2mp MBONs, and KCs-γ3, γ3β'1 MBONs connections. Consequently, changing these synaptic connections requires involving more training sessions.

## Materials and methods

### Fly strains

All flies (*Drosophila melanogaster*) were raised on standard cornmeal medium at 23℃ and 60% relative humidity under a 12 hr light-dark cycle as described (*Shuai et al., 2010*). The control strain was $w^{1118}$(isoCJ1). UAS-Shibire$^{ts1}$ and UAS-Kir2.1;tub-gal80$^{ts}$ were extant stocks in the laboratory. Gal4 lines used have been described previously: *TH-Gal4* (*Friggi-Grelin et al., 2003*), *C739-Gal4* (*McGuire et al., 2001*), *R58E02-Gal4, R52H03-Gal4, R13F02-Gal4,* and *R71D08-Gal4* (*Jenett et al., 2012*), *5HT1b-Gal4* (*Yuan et al., 2005*), *G0239-Gal4* (*Pai et al., 2013*), and *VT30604-Gal4* (#200228) (*Wu et al., 2013*). *UAS-GCaMP7f* was a gift from Yi Sun, Westlake University.

### Behavioral assays

All flies were raised at 23℃ and mixed-sex populations of 2- to 5-day-old flies were used in all experiments. The Pavlovian olfactory aversive conditioning procedure was performed as described previously in a behavioral room at 23℃ and 60% relative humidity described (*Shuai et al., 2010*).

During training, roughly 80 flies successfully received the following stimuli in a training tube, which contained a copper grid: air for 90 s, an odor paired with 12 pulses of 60 V electric shock (CS +) for 1 min, air for 45 s, a second odor without pairing the electric shock (CS–) for 60 s, and finally air for 45 s. 3-Octanol (OCT, 10–15 µl in 10 ml mineral oil), 4-methylcyclohexanol (MCH, 10 µl in 10 ml mineral oil), and ethyl acetate (EA, 10 µl in 10 ml mineral oil) were used as standard odorants. This process describes a single training trial. Six sequential trials with 15 min intervals constitute spaced training. For memory retrieval, trained flies were transferred into a T-maze, where they were allowed 2 min to choose between two odors (CS+ and novel odor, or CS– and novel odor).

Odor performance was quantified by a PI (performance index) calculated based on the fraction of flies in the two T-maze arms. A PI of 100 indicated that all flies avoid the odor exposed during training (either CS+ or CS–), while a PI of 0 indicated no odor preference, as reflected by 50:50 distributions between the arms. To balance naïve odor bias, two reciprocal groups were trained and tested simultaneously and the testing odors were always OCT and MCH, one serves as the CS+ or CS– and the other serves as the novel odor alternatively. One complete PI was generated from the average PI of a couple of rotate training and testing. Specifically, for the testing of CS+ performance, either OCT or MCH was alternately used as CS+, and EA was CS–; whereas for the testing of CS– performance, either OCT or MCH was alternately used as CS–, and EA was CS+. For 3 min memory, flies were tested immediately after training. For longer memory retention, flies were placed in a fresh vial with the same contents as the vial they had been kept in before the training until the test.

For odor avoidance testing, a similar PI was calculated based on the fraction of flies in the two T-maze arms. A PI of 100 indicated that all flies choose the air arm, while a PI of 0 indicated no preference.

For cold-shock anesthesia, flies were transferred into pre-chilled plastic vials and kept on ice for 2 min as described previously (*Shuai et al., 2010*).

For cycloheximide (CXM) feeding, flies were provided food with (CXM+) or without (CXM–) 35 mM CXM (Sigma-Aldrich) dissolved in control solution, and 5% (wt/vol) glucose, for 16 hr before and after training until memory retention was tested.

For neural inactivation experiments using *UAS-Shi$^{ts}$*, crosses were reared at 23℃ to avoid unintended inactivation. Flies were shifted to 32℃ 15 min in advance for neural inhibition during training or test.

For odor re-exposure, flies were exposed to three cycles of conditioned odor (either CS+ or CS–) for 1 min with 1 min interval as described previously (*Wang et al., 2019*). All mentioned re-exposure treatments were performed immediately before testing.

## Immunohistochemistry

Flies were quickly anesthetized on ice and whole brains were dissected in ice-cold PBS within 5 min, then stained as described (*Shuai et al., 2010*). Brains were fixed in 4% paraformaldehyde in PBS for 30 min on ice. Brains were incubated for at least 72 hr with the primary antibodies anti-GFP (chicken, 1:2000; Abcam, Cambridge, UK), anti-nc82 (mouse, 1:10; Developmental Studies Hybridoma Bank, Iowa City, IA), and anti-DsRed (rabbit, 1:500; Takara Bio, Kyoto, Japan). Brains were washed three times again in PBS with 0.2% Triton X-100 and transferred into secondary antibody solution (anti-chicken Alexa Fluor 488, 1:200; anti-mouse Alexa Fluor 647, 1:200; anti-rabbit Alexa Fluor 647, 1:200; Molecular Probes, Eugene, OR) and incubated for 48 hr at 4˚C. Images were obtained using a Zeiss LSM710 confocal microscope (Carl Zeiss AG, Oberkochen, Germany).

## In vivo two-photon calcium imaging

Two- to five-day-old female flies were used. After anesthetized in a plastic vial on ice for 15–20 s, flies were then gently inserted into a hole of a thin plastic rectangular plate and stabilized in the hole by glue. In a saline bath, the area surrounding the region of interest was surgically removed to expose the dorsal side of the brain. Fat and air sacs were gently removed to give a clear view of the brain. For calcium response imaging, the 40× water immersion objective lens (NA=1.0; Zeiss) were lowered near the exposed brain.

Imaging was performed on a Zeiss LSM 7MP two-photon laser scanning microscope with an imaging wavelength at 910 nm (Carl Zeiss AG). The $512 \times 512$ pixel images were acquired at 2.6 Hz. In each trial, 10 s of baseline was recorded, followed by an odor-evoked response recording. Two to three minutes of rest were given in between trials when multiple trials were applied. GCaMP responses and the average value within 10 s before and during stimulus were quantified using custom software written in MatLab (MathWorks, Natick, MA). For the brain region of interest during the experimental period, the average fluorescence value, $F_{ave}$, was then converted to $\Delta F/F_0$ using the formula $\Delta F/F_0 = (F_{ave}-F_0)/F_0$, where $F_0$ is the baseline fluorescence value, measured as the average of 10 s before the stimulus. Flies were trained using CS+ (OCT, 10 µl in 10 ml mineral oil) and CS– (MCH 10 µl in 10 ml mineral oil), and IA (10 µl in 10 ml mineral oil) was used as novel odor. To gain the calibrated calcium responses of trained flies, the ratios of $\Delta F/F_0$ to different odors of trained and naïve flies were calculated. Specifically, all odor-evoked responses after training were divided to the mean responses to the same odors of same number naïve flies. Thus, a calibrated $\Delta F/F_0$ of 1.0 indicated no difference in the odor-evoked responses to the correspondence odor between trained and naïve group.

## Statistics

Statistics were performed with GraphPad Prism software (version 7; GraphPad Software, San Diego, CA). All data satisfied the assumption of normal distribution (one-sample Kolmogorov–Smirnov test). Comparisons between two groups were performed using two-tailed t-tests. Comparisons of multiple groups were performed using one-way or two-way analysis of variance (ANOVA) tests followed by Bonferroni correction for multiple comparisons. p-Values less than 0.05 were considered statistically significant and are marked with an asterisk in figures; NS indicates non-significant differences ($p > 0.05$).

## Acknowledgements

We thank BDSC and VDRC, and Dr Yi Sun for fly stocks. We thank Bo Lei, Yijun Niu, Dr Xuchen Zhang, Dr Wantong Hu, Dr Dongqin Cai, Dr Yunchuan Zhang, Ning Huang, Shiqiang Hu, Yikai Tang, Shunan Wang, and Dr Stella Christie for helpful discussions; Bowen Chen for software supply; Lianzhang Wang for facility supply. This work was supported by grants from the National Science Foundation of China (31970955, to QL), and the Tsinghua-Peking Center for Life Sciences.

## Additional information

### Funding

| Funder | Grant reference number | Author |
|---|---|---|
| National Science Foundation of China | 31970955 | Qian Li |
| Tsinghua-Peking Center for Life Sciences | | Yi Zhong |

The funders had no role in study design, data collection and interpretation, or the decision to submit the work for publication.

### Author contributions

Bohan Zhao, Conceptualization, Formal analysis, Validation, Investigation, Visualization, Methodology, Writing - original draft, Project administration, Writing - review and editing, Behavioral Experiments; Jiameng Sun, Formal analysis, Investigation, Calcium Imaging and Immunostaining; Qian Li, Formal analysis, Funding acquisition, Writing - original draft; Yi Zhong, Conceptualization, Resources, Formal analysis, Supervision, Funding acquisition, Writing - original draft, Writing - review and editing

### Author ORCIDs

Bohan Zhao (iD) https://orcid.org/0000-0002-9177-1278
Yi Zhong (iD) https://orcid.org/0000-0002-7927-5976

### Decision letter and Author response

Decision letter https://doi.org/10.7554/eLife.66499.sa1
Author response https://doi.org/10.7554/eLife.66499.sa2

## Additional files

### Supplementary files

- Source code 1.
- Transparent reporting form

### Data availability

All data generated or analysed during this study are included in the manuscript and supporting files.

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
