## [Decision Letter]

**Acceptance summary:**

This work reveals a novel form of *Drosophila* long-term memory that is of potential interest to most neuroscientists working on various animals. Classical protein-synthesis-dependent long-term memory forms only after repetitive spaced trials of olfactory conditioning. The authors discovered that flies also form a "blurred" or "vague" protein-synthesis-dependent long-term memory which distinguishes experienced two odors from a third naive odor after single-trial training. This merged long-term memory lacking the event details likely occurs in most animals since long-lasting memory of occasional threatening experiences for future escape behavior is crucial for survival.

**Decision letter after peer review:**

Thank you for submitting your article "Differential Conditioning Produces Merged Long-term Memory in *Drosophila*" for consideration by *eLife*. Your article has been reviewed by 2 peer reviewers, and the evaluation has been overseen by K VijayRaghavan as the Senior and Reviewing Editor. The following individual involved in review of your submission has agreed to reveal their identity: Ann-Shyn Chiang (Reviewer #1).

Essential revisions:

The authors should keep these essential revisions in mind while attending to the reviewers' comments which are appended below.

The final model presented in the Zhong manuscript presents an integrated view of the results obtained by them and by Jacob and Wadell, Neuron 2020. To do this it is important to replicate some of the essential results from Jacob et al. In addition the reviewers put a high value on manuscript data that replicates data, in the same or complementary ways from other laboratories. Please keep these points in mind while preparing your revisions.

1. By using cell-type-specific Gal4 lines that are readily available, target neurons for "the avoiding CS+ memory", "the early approaching CS- memory" and "the late avoiding CS- memory" should be identified. For example, PPL1-DANs and PAM-DANs expressed in TH-Gal4 can be respectively tested with specific split-Gal4 lines.

2. In Figure 2B, when CS- was presented 20min after CS+ even the mLTM of CS+ vs novel odor was impaired, the result means that the formation of CS+ mLTM requires the presence of CS- within a certain time window. This raises an alternative interpretation of the data that CS- presented 20min after CS+ actually serves as interference for the formation of CS+ mLTM, rather than the merged CS+/CS- memory. As to distinguish these two hypotheses, training the flies with only CS+ without delivering CS-, and then test between CS+ and novel odor should be able to address the question. If presenting CS+ alone fails to form LTM of CS+ vs the third odor, then CS- presented within a certain time window is necessary. Otherwise, CS- presented 20min after CS- may serve as an interference.

3. In Figure 4B, the author measured the odor responses of CS+, CS-, and novel odor in MBON-a2sc one day after association, and found that both CS+ and CS- showed depression response. To strengthen the link between the functional response of MBON-a2sc and behavioral phenotype, conditions abolished mLTM by reactivation or prolong odor delivery interval (Figure 2A and 2B) should also be tested to see whether the memory trace are also abolished. Also, what is the responses immediately after 1X training? As CS- mLTM changes from appetitive to avoidance within the first 24 hours after training, the calcium responses of MBON-a2 during this period are necessary for understanding the circuit mechanism of mLTM dynamics.

4. (Point 1 of Reviewer 2) The observation that presentation of a novel third odor leads to mLTM after only a single session of aversive conditioning is intriguing. Authors describe in their methods using three odors for their experiments (as CS+, CS- or novel), but did not alternate/rotate the different combination pairings used as the "novel" one. A panel of odors as "novel", not listed in the manuscript, should be tested which will strengthen the larger conceptual framework and impact. In addition, the authors should perform at least a subset of the experiment using air during testing rather than a 3rd odor.

5. (Point6 of Reviewer 2) More experimentation and discussion regarding the differences between single-trial conditioning to form mLTM and spaced conditioning to form complementary LTM, is required. The authors contrast/merge their behavioral results with those published by Jacob et al., (2020). The authors should reproduce the essence of those found by Jacob et al., and publish them in this paper. Replication of experimental results across labs is very important, especially for behavioral outcomes and when models are constructed using results obtained by other investigators. The authors allude to the two pairs of DAN that project to α2sc MBN for this plasticity, but did not specifically mention those DAN (lines 220-221) nor elaborate on this speculation.

*Reviewer #1 (Recommendations for the authors):*

The finding of mLTM opens many questions. Knowing it is impractical to address all these questions in one study, I thus suggest only essential experiments to strengthen the report.

1. By using cell-type specific Gal4 lines that are readily available, target neurons for "the avoiding CS+ memory", "the early approaching CS- memory" and "the late avoiding CS- memory" should be identified. For example, PPL1-DANs and PAM-DANs expressed in TH-Gal4 can be respectively tested with specific split-Gal4 lines.

2. In Figure 2B, when CS- was presented 20min after CS+ even the mLTM of CS+ vs novel odor was impaired, the result means that the formation of CS+ mLTM requires the present of CS- within certain time window. This raises an alternative interpretation of the data that CS- presented 20min after CS+ actually serves as an interference for the formation of CS+ mLTM, rather than the merged CS+/CS- memory. As to distinguish these two hypothesis, training the flies with only CS+ without delivering CS-, and then test between CS+ and novel odor should be able to address the question. If presenting CS+ alone fails to form LTM of CS+ vs the third odor, then CS- presented within certain time window is necessary. Otherwise, CS- presented 20min after CS- may serve as an interference.

3. In Figure 4B, the author measured the odor responses of CS+, CS-, and novel odor in MBON-a2sc one day after association, and found that both CS+ and CS- showed depression response. To strengthen the link between functional response of MBON-a2sc and behavior phenotype, conditions abolished mLTM by reactivation or prolong odor delivery interval (Figure 2A and 2B) should also be tested to see whether the memory trace are also abolished. Also, what is the responses immediately after 1X training? As CS- mLTM changes from appetitive to avoidance within the first 24 hours after training, the calcium responses of MBON-a2 during this period are necessary for understanding the circuit mechanism of mLTM dynamics.

In summary, the discovery of mLTM is interesting. However, to sustain the authors' claims that the same neurons are involved in both CS+/CS-avoidance, the authors need to use specific split-Gal4 lines expressed in only the target neurons. Also, to claim this is a merged LTM, authors need to address how the trained flies turn from the initial CS- approaching memory to late CS- avoidance memory. Without additional data to address these critical issues, the authors should largely tune down their claims.

*Reviewer #2 (Recommendations for the authors):*

1) Citations or references are not included when the phrase "described previously" is used (lines 238, 248, 275, 287).

2) Provide more detail on odor re-exposure methods (lines 283-284). When was this administered?

3) Explain with more detail by "calibrated calcium responses" in methods (lines 316-318).

4) Include sample size used for experiments. It is difficult to count individual data points when variability of data is tight.

5) Figure 3A cartoon – α'3 lobe ( α='3 tip) should be in magenta – innervated by PPL1 DAN.

6) suggest "unconditioned stimulus" rather than "non-conditioned stimulus"

7) organization of references when listed in the text – suggest organizing them chronologically or alphabetically. Currently there is random organization.

8) Figure 2A – why use the term "reactivation" rather than "extinction trials?"

9) The Abstract and the Introduction use the extremely vague phrase, "…that animals utilize escape strategies for facing occasional and repeated dangers…" Either bring this point home for the reader with specific and concrete examples in the Discussion or modify/eliminate the sentence.

10) A good overhaul of the referencing throughout the manuscript is warranted. One glaring omission of perhaps several references that should be offered is in support of the sentence, "In *Drosophila*, aversive olfactory memories are mainly encoded by the PPL1 cluster of DANs," (line 124). Schwaerzel et al., 2003, and others.

---

## [Author Response]

Essential revisions:The authors should keep these essential revisions in mind while attending to the reviewers' comments which are appended below.The final model presented in the Zhong manuscript presents an integrated view of the results obtained by them and by Jacob and Wadell, Neuron 2020. To do this it is important to replicate some of the essential results from Jacob et al. In addition the reviewers put a high value on manuscript data that replicates data, in the same or complementary ways from other laboratories. Please keep these points in mind while preparing your revisions.1. By using cell-type-specific Gal4 lines that are readily available, target neurons for "the avoiding CS+ memory", "the early approaching CS- memory" and "the late avoiding CS- memory" should be identified. For example, PPL1-DANs and PAM-DANs expressed in TH-Gal4 can be respectively tested with specific split-Gal4 lines.

We thank the reviewer for this suggestion. In the revised manuscript, we added the suggested experiments with two more specific Gal4 lines, R58E02 for PAM-DANs and R52H03 for PPL1-DANs (Figure 3—figure supplement 1A, B). The 24-h avoidances to CS+ and CS– were both impaired by blocking PPL1-, but not PAM-DANs (Figure 3A). Interestingly, although the early avoidance to CS+ was impaired specifically by blocking PPL1-DANs, the immediate response to CS– was affected oppositely when PPL1-DANs or PAM-DANs were blocked (Figure 3—figure supplement 2). These data support our theoretical model in the revised edition that the response to CS– is the sum of two opposite memory components: the hours-lasting safety-memory to CS– and the aversive mLTM (Figure 6B).

2. In Figure 2B, when CS- was presented 20min after CS+ even the mLTM of CS+ vs novel odor was impaired, the result means that the formation of CS+ mLTM requires the presence of CS- within a certain time window. This raises an alternative interpretation of the data that CS- presented 20min after CS+ actually serves as interference for the formation of CS+ mLTM, rather than the merged CS+/CS- memory. As to distinguish these two hypotheses, training the flies with only CS+ without delivering CS-, and then test between CS+ and novel odor should be able to address the question. If presenting CS+ alone fails to form LTM of CS+ vs the third odor, then CS- presented within a certain time window is necessary. Otherwise, CS- presented 20min after CS- may serve as an interference.

As suggested by the reviewer, we have performed new experiments with absolute training (training without CS– or CS+) in the revision, and such training failed to induce 24-h avoidances (Figure 2—figure supplement 1). The obtained data support presenting CS+ and CS– are both necessary for forming mLTM. Moreover, this conclusion can be further strengthened by the result that mLTM was gradually decreased with the prolonged intervals between CS+ and CS– (Figure 2B).

3. In Figure 4B, the author measured the odor responses of CS+, CS-, and novel odor in MBON-a2sc one day after association, and found that both CS+ and CS- showed depression response. To strengthen the link between the functional response of MBON-a2sc and behavioral phenotype, conditions abolished mLTM by reactivation or prolong odor delivery interval (Figure 2A and 2B) should also be tested to see whether the memory trace are also abolished. Also, what is the responses immediately after 1X training? As CS- mLTM changes from appetitive to avoidance within the first 24 hours after training, the calcium responses of MBON-a2 during this period are necessary for understanding the circuit mechanism of mLTM dynamics.

We provide new data in revision that training with 10-min temporal intervals fails to form the long-term depression in α2sc MBONs (Figure 5). In addition, we found that the depression of odor-evoked responses to CS+ and CS– can be observed immediately after training (Figure 5—figure supplement 1A). Our data suggest that the merged memory should be encoded during training (Figure 5—figure supplement 1B) and then consolidated within 8 hours (Figure 1D).

4. (Point 1 of Reviewer 2) The observation that presentation of a novel third odor leads to mLTM after only a single session of aversive conditioning is intriguing. Authors describe in their methods using three odors for their experiments (as CS+, CS- or novel), but did not alternate/rotate the different combination pairings used as the "novel" one. A panel of odors as "novel", not listed in the manuscript, should be tested which will strengthen the larger conceptual framework and impact. In addition, the authors should perform at least a subset of the experiment using air during testing rather than a 3rd odor.

We appreciate that the reviewer pointed out our mistake. We added a detailed description of the odor combination protocol in the Methods (line 244-257). A panel of odors, OCT and MCH, were used as the novel odor alternatively throughout our study. In addition, we performed the suggested experiment and observed increased 24-h avoidance to CS+ and CS– when they are compared with air (Figure 1—figure supplement 2), which is consistent with our findings.

5. (Point6 of Reviewer 2) More experimentation and discussion regarding the differences between single-trial conditioning to form mLTM and spaced conditioning to form complementary LTM, is required. The authors contrast/merge their behavioral results with those published by Jacob et al., (2020). The authors should reproduce the essence of those found by Jacob et al., and publish them in this paper. Replication of experimental results across labs is very important, especially for behavioral outcomes and when models are constructed using results obtained by other investigators. The authors allude to the two pairs of DAN that project to α2sc MBN for this plasticity, but did not specifically mention those DAN (lines 220-221) nor elaborate on this speculation.

We thank the reviewer for this suggestion and replicated the essential result of Jacob et al., (2020). Consistently, we observed complementary 24-h memories were formed after repetitive spaced training (Figure 1—figure supplement 1). Moreover, as suggested by the reviewer, we added more discussion of the differences between mLTM and complementary LTMs, and potential neural mechanisms (line 239-249).

Reviewer #1 (Recommendations for the authors):The finding of mLTM opens many questions. Knowing it is impractical to address all these questions in one study, I thus suggest only essential experiments to strengthen the report.1. By using cell-type specific Gal4 lines that are readily available, target neurons for "the avoiding CS+ memory", "the early approaching CS- memory" and "the late avoiding CS- memory" should be identified. For example, PPL1-DANs and PAM-DANs expressed in TH-Gal4 can be respectively tested with specific split-Gal4 lines.

We thank the reviewer for this suggestion. In the revised manuscript, we added the suggested experiments with two more specific Gal4 lines, R58E02 for PAM-DANs and R52H03 for PPL1-DANs (Figure 3—figure supplement 1A, B). The 24-h avoidances to CS+ and CS– were both impaired by blocking PPL1-, but not PAM-DANs (Figure 3A). Interestingly, although the early avoidance to CS+ was impaired specifically by blocking PPL1-DANs, the immediate response to CS– was affected oppositely when PPL1-DANs or PAM-DANs were blocked (Figure 3—figure supplement 2). These data support our theoretical model in the revised edition that the response to CS– is the sum of two opposite memory components: the hours-lasting safety-memory to CS– and the aversive mLTM (Figure 6B).

2. In Figure 2B, when CS- was presented 20min after CS+ even the mLTM of CS+ vs novel odor was impaired, the result means that the formation of CS+ mLTM requires the present of CS- within certain time window. This raises an alternative interpretation of the data that CS- presented 20min after CS+ actually serves as an interference for the formation of CS+ mLTM, rather than the merged CS+/CS- memory. As to distinguish these two hypothesis, training the flies with only CS+ without delivering CS-, and then test between CS+ and novel odor should be able to address the question. If presenting CS+ alone fails to form LTM of CS+ vs the third odor, then CS- presented within certain time window is necessary. Otherwise, CS- presented 20min after CS- may serve as an interference.

As suggested by the reviewer, we have performed new experiments with absolute training (training without CS– or CS+) in the revision, and such training failed to induce 24-h avoidances (Figure 2—figure supplement 1). The obtained data support presenting CS+ and CS– are both necessary for forming mLTM. Moreover, this conclusion can be further strengthened by the result that mLTM was gradually decreased with the prolonged intervals between CS+ and CS– (Figure 2B).

3. In Figure 4B, the author measured the odor responses of CS+, CS-, and novel odor in MBON-a2sc one day after association, and found that both CS+ and CS- showed depression response. To strengthen the link between functional response of MBON-a2sc and behavior phenotype, conditions abolished mLTM by reactivation or prolong odor delivery interval (Figure 2A and 2B) should also be tested to see whether the memory trace are also abolished. Also, what is the responses immediately after 1X training? As CS- mLTM changes from appetitive to avoidance within the first 24 hours after training, the calcium responses of MBON-a2 during this period are necessary for understanding the circuit mechanism of mLTM dynamics.

We provide new data in revision that training with 10-min temporal intervals fails to form the long-term depression in α2sc MBONs (Figure 5). In addition, we found that the depression of odor-evoked responses to CS+ and CS– can be observed immediately after training (Figure 5—figure supplement 1A). Our data suggest that the merged memory should be encoded during training (Figure 5—figure supplement 1B) and then consolidated within 8 hours (Figure 1D).

Reviewer #2 (Recommendations for the authors):1) Citations or references are not included when the phrase "described previously" is used (lines 238, 248, 275, 287).2) Provide more detail on odor re-exposure methods (lines 283-284). When was this administered?3) Explain with more detail by "calibrated calcium responses" in methods (lines 316-318).4) Include sample size used for experiments. It is difficult to count individual data points when variability of data is tight.5) Figure 3A cartoon – α'3 lobe ( α='3 tip) should be in magenta – innervated by PPL1 DAN.

We thank the reviewer for these helpful suggestions and modified the text and figures accordingly.

6) suggest "unconditioned stimulus" rather than "non-conditioned stimulus"

Considering the “unconditioned stimulus” has been widely defined as the reward or the punishment (in this case, the electric shock) in memory studies, here we used the “non-conditioned stimulus” to indicate the CS–.

7) organization of references when listed in the text – suggest organizing them chronologically or alphabetically. Currently there is random organization.

The list of references was organized alphabetically based on the first author’s last name in the manuscript as the requirement of *eLife*.

8) Figure 2A – why use the term "reactivation" rather than "extinction trials?"

We modified the figure accordingly in revision (Figure 2A).

9) The Abstract and the Introduction use the extremely vague phrase, "…that animals utilize escape strategies for facing occasional and repeated dangers…" Either bring this point home for the reader with specific and concrete examples in the Discussion or modify/eliminate the sentence.

As the reviewer suggested, we added examples in the revised *Discussion* (line 233-237).

10) A good overhaul of the referencing throughout the manuscript is warranted. One glaring omission of perhaps several references that should be offered is in support of the sentence, "In *Drosophila*, aversive olfactory memories are mainly encoded by the PPL1 cluster of DANs," (line 124). Schwaerzel et al., 2003, and others.

We appreciate that the reviewer pointed out our mistake. We updated references in revision.